# How N-Acetylcysteine Supplementation Affects Redox Regulation, Especially at Mitohormesis and Sarcohormesis Level: Current Perspective

**DOI:** 10.3390/antiox10020153

**Published:** 2021-01-21

**Authors:** Aslı Devrim-Lanpir, Lee Hill, Beat Knechtle

**Affiliations:** 1Department of Nutrition and Dietetics, Faculty of Health Sciences, Istanbul Medeniyet University, Istanbul 34862, Turkey; asli.devrim@medeniyet.edu.tr; 2Division of Gastroenterology and Nutrition, Department of Pediatrics, McMaster University, Hamilton, ON L8S 4K1, Canada; hilll14@mcmaster.ca; 3Medbase St. Gallen am Vadianplatz, 9001 St. Gallen, Switzerland; 4Institute of Primary Care, University of Zurich, 8091 Zurich, Switzerland

**Keywords:** N-acetylcysteine, mitochondrial adaptation, skeletal adaptation, hormesis, oxidative stress

## Abstract

Exercise frequently alters the metabolic processes of oxidative metabolism in athletes, including exposure to extreme reactive oxygen species impairing exercise performance. Therefore, both researchers and athletes have been consistently investigating the possible strategies to improve metabolic adaptations to exercise-induced oxidative stress. N-acetylcysteine (NAC) has been applied as a therapeutic agent in treating many diseases in humans due to its precursory role in the production of hepatic glutathione, a natural antioxidant. Several studies have investigated NAC’s possible therapeutic role in oxidative metabolism and adaptive response to exercise in the athletic population. However, still conflicting questions regarding NAC supplementation need to be clarified. This narrative review aims to re-evaluate the metabolic effects of NAC on exercise-induced oxidative stress and adaptive response developed by athletes against the exercise, especially mitohormetic and sarcohormetic response.

## 1. Introduction

Exercise is defined as a complex challenge for the body’s adaptive mechanism and homeostatic balance [1]. The body responds to exercise-induced oxidative stress by (a) eliminating oxidants through the removal of oxidants or by upregulating antioxidant concentrations; (b) reducing the concentration of oxidants at baseline; (c) increasing muscle regeneration and remodelling capacity in damaged muscles after exercise; and (d) upregulating the anti-oxidant defence system through numerous adaptive transcriptional genes, thus trying to adapt for further challenging oxidative processes [2]. However, the endogenous response to exercise varies according to both the duration and the intensity of the training [3]. While mild to moderate exercise training has extensive effects on the body’s hormetic balance by upregulating exercise-dependent gene regulation and leading to exercise adaptations. Intensified or prolonged exercise training leads to an excessive accumulation of reactive oxygen species (ROS) by exceeding the body’s endogenous antioxidant capacity, thus leading to inflammatory, oxidative, physiological, immunological, and neuroendocrinological disorders [4]. These disorders then lead to a deterioration in sports performance and impari athletic success [1].

An important aspect of physiological and metabolic adaption to ROS is an increased body tolerance to exercise-induced progressive oxidant accumulation and is known as “hormesis theory” [5]. Hormesis is represented by a J- or bell-shaped curve indicating that too low or excessive ROS accumulation leads to deleterious changes, whereas mild to moderate ROS is essential for metabolic and physiological processes [6]. However, an exercise-induced excessive oxidant accumulation causes both mitochondrial and skeletal oxidative damage by damaging cellular proteins and lipids in mitochondria [6] and contractile myocytes [7], and more seriously altering multiple cellular signal cascades related to cellular function, adaptation, and homeostasis [4]. Therefore, it is important to maintain an optimum relationship between the dose of ROS and body response to maintain both mitochondrial and skeletal homeostasis.

As ROS can be a highly compelling factor for sports performance and metabolic regulations [4], exogenous antioxidant supplementation has been extensively studied to investigate their effectiveness in maintaining redox homeostasis in states of exercise-induced oxidative stress in the athletic population [8,9,10,11]. However, the consequences of antioxidant supplementation against high ROS have yielded controversial results, including altering exercise-induced gene expression resulting in decreased or blunted physiological adaptation [12]. To define an antioxidant supplement as effective, it should lower the exercise-induced oxidative stress by maintaining and promoting the body’s adaptive response to exercise. N-acetylcysteine (NAC) is considered as one of the most promising antioxidant substances compared to other antioxidant supplements such as glutathione, and vitamin E and C [13]. Glutathione (GSH) is one of the most effective endogenous antioxidants that can be synthesized primarily in the liver [14]. In addition, it can be obtained exogenously from diet or supplements [15]. NAC, which is a precursor of glutathione, provides the maintenance of the glutathione synthesis by releasing cysteine, a rate-limiting protein for this reaction [16]. However, although NAC supplementation looks promising in exogenous antioxidants and is, therefore, a common belief among trainers and athletes that it has great performance benefits [17], the effect of NAC on hormesis appears to be variable regarding literature findings.

This narrative review will discuss the effects of NAC supplementation on the hormetic response to exercise. We identified all studies on N-acetylcysteine, reactive oxygen species, and hormesis using the PubMed, Cochrane Library, Web of Science databases by publication applied in human subjects, searching Mesh terms (“Acetylcysteine/adverse effects”(Mesh) OR “Acetylcysteine/immunology”(Mesh) OR “Acetylcysteine/metabolism”(Mesh) OR “Acetylcysteine/supply and distribution”(Mesh) OR “Acetylcysteine/therapeutic use”(Mesh) OR “Reactive Oxygen Species/metabolism”(Mesh) OR “Reactive Oxygen Species/blood”(Mesh) OR “Reactive Oxygen Species / adverse effects”(Mesh) OR “Reactive Oxygen Species/immunology”(Mesh) OR “Oxidative Stress/immunology”(Mesh) OR “Oxidative Stress/physiology”, including below this term in the Mesh hierarchy such as “Protein Carbonylation”, “Nitrosative Stress”, “Hydroxyl Radical”, “Peroxides”, “Hydrogen Peroxide”, “Lipid Peroxides”, “Superoxides”, “Nitric Oxide”). We have used other entry words including “N-Acetyl-L-cysteine”, “N Acetyl L cysteine”, “N-Acetylcysteine”, “N Acetylcysteine”, “Oxygen Species, Reactive”, “Active Oxygen”, “Oxygen, Active”, “Oxygen Radicals”, “Pro-Oxidants”, “Pro Oxidants”. We included articles published until November 2020 (Table 1). Original articles (i.e., research papers and case reports) on N-acetylcysteine, reactive oxygen species, and hormesis were considered eligible for the review.

## 2. Mitochondrial Adaptation to Exercise and Training: Mitohormesis

Mitochondria are highly dynamic organelles in which ROS are commonly produced during various processes such as dispersion of the electron transport chain and membrane potential by ATP synthase [18]. It is well-known that ROS production is inevitable during oxidative phosphorylation [19]. Any ROS production from mild to excess causes changes in mitochondrial function [20]. Due to the close integration of mitochondria into the cell, mitochondrial changes are immediately transmitted to the cytosol and nucleus, inducing nuclear transcription and cell signalling pathways [6]. Mitochondrial ROS modulation is regulated in a dose-dependent manner [21]. For instance, low ROS concentration initiates a mitochondrial stress signal cascade, a mitochondrial attempt to compensate and adopt exercise-induced stress changes [21]. All these cytoprotective adaptations occurring in the mitochondria are called mitohormesis [6].

Mitohormesis involves an increasing number of mitochondria, an increased endogenous antioxidant response termed mitochondrial biogenesis, and an altered cytoprotective gene expression [18]. It is dynamically regulated by processes of fusion, fusion and mitophagy [22]. While mitochondrial fission is necessary to create new mitochondria, eliminate damaged mitochondria and facilitate apoptosis during extreme cellular stress, mitochondrial fusion provides mitochondrial networks for exchanging DNA and several metabolites between highly interconnected mitochondrial tubules, thereby allowing the preservation of cellular function by allowing mitochondrial DNA (mtDNA) mutations [22].

Many mitochondrial redox-sensitive transcription factors act to modulate the antioxidant response, including mitochondrial topoisomerase I (Top1mt) [23], AMP-activated protein kinase (AMPK) [24], and Peroxisome proliferator-activated receptor gamma coactivator 1-alpha (PGC-1a) [25]. Studies revealed that increased mitochondrial ROS activates multiple redox-sensitive kinases, including AMPK and the Mitogen-activated protein kinases (MAPK p38) [26], c-Jun N-terminal kinase (JNK) [27], and extracellular signal-regulated kinases (ERK1/2) [28], which regulate mitochondrial biogenesis and mitohormesis modulation through PGC1-a [25]. Phosphorylated PGC-1a by redox-sensitive kinases modulates numerous muscle-related transcriptional factors through nuclear respiratory factors 1 and 2 (NRF-1 and NRF-2) [29], cAMP Response Element-Binding Protein (CREB) [30], Histone deacetylase 1 (HDAC1) [31], mitochondrial transcription factor A (mtTFA) [32], and myocyte enhancer factor-2 (MEF2) [33], including improving muscle metabolism and enhancing muscle performance. PGC1-a also regulates mitochondrial metabolism and biogenesis by activating mitochondrial transcriptional genes such as mitofusin 1/2 (Mfn1/2), cytochrome C, cytochrome oxidase, and various genes involved in the tricarboxylic acid cycle and oxidative phosphorylation process [34]. Additionally, as PGC1a enhances antioxidant enzymes such as superoxide dismutase (SOD), glutathione peroxidase (GPx) and catalase, mitochondrial biogenesis actively participates in redox homeostasis by regulating antioxidant enzymes [34]. Other factors are mitochondria-generated cyclic AMP (cAMP), which play a substantial role in the reversible phosphorylation, and sirtuins that mediate deacetylation processes within the mitochondria including activation of protein kinase A, a regulator of ATP production, also referring to as the main mitochondrial regulators [35].

Mitohormesis affects cell longevity by triggering cytoprotective mitochondrial signal pathways, increasing oxidative stress resistance and insulin sensitivity, and thus prolonging life span [6,18]. Excessive ROS accumulation causes mitochondrial damage, resulting in mitochondrial fission and mitophagy [36]. In the case when the endogenous or exogenous ROS response is insufficient, these damages may even lead to mitoptosis or cell death [36]. Therefore, maintaining mitohormesis is of paramount importance for both performance and health.

## 3. Skeletal Muscle Adaptation to Exercise and Training: Sarcohormesis

Exercise-dependent ROS are produced in multiple locations in muscle cells [37]. Mitochondria are considered to be one of the primary sources of reactive oxygen production in contractile muscles and it has been estimated that 2% of the oxygen consumed by mitochondria undergoes single-electron reduction, resulting in oxygen conversion to superoxide [38]. Therefore, it is assumed that many mechanisms are involved in the production of ROS in skeletal muscle.

Exercise-mediated ROS create an enormous mechanical overload on skeletal muscles, resulting in an anti-inflammatory response orchestrated by neutrophils and induce adaptive signaling pathways within the muscle [39]. Additionally, numerous myokines such as interleukin (IL)-6 and IL-10 start to be synthesized in response to exercise-mediated adaptive and hypertrophic signals [40]. Strenuous/prolonged exercise, in particular, if it includes eccentric exercise, leads to a dramatically increased ROS generation mainly by NOX, xanthine oxidase and phospholipase A2 enzymes and several mitochondrial processes, causing reversible and irreversible exercise-induced muscle damage, soreness, microtrauma, and secondary damage of uninjured myofibers [41]. In these conditions, the survival of skeletal muscle is mainly dependent on its ability to regenerate and remodel in response to metabolic changes caused by exercise [3]. Pro- and anti-inflammatory macrophages are well-known for their predominant influence in skeletal muscle proliferation, remodelling and redox control by activating the myogenic response of skeletal muscle cells after exercise [42]. The myogenic response is commonly orchestrated by several muscle cells, including myosatellite cells, myofibers, fibroadipogenic progenitors, and endothelial cells [43].

Although the exercise-induced ROS production seems to be detrimental to muscle function in the short term, it reveals beneficial effects in the long term by triggering the adaptive metabolic and nuclear response of muscle cells [37]. Mild stress induced by exercise rapidly up-regulate cell signalling pathways that alter nuclear gene expression in skeletal muscle, resulting in cytoprotective adaptations, termed sarcohormesis [4,44].

The effect of exercise-induced ROS on skeletal muscle and sarcohormesis response depends on the state of several internal and external factors, including the amount of ROS targeting muscle cells and the duration of ROS exposure, endogenous antioxidant status, and myogenic capacity of target muscle cells [45].

The amount of ROS produced during exercise is one of the main determinants of the muscle’s hormetic response [46]. Low concentrations of ROS in muscle cells activated either by direct exposure to ROS or by changes in the redox balance of thiols have a positive effect by increasing Ca^2+^ release from the sarcoplasmic reticulum (SR) to increase force generation in skeletal muscle [7]. However, overexposure to ROS during prolonged or strenuous exercise causes dysregulation of myofibrillar Ca^2+^ sensitivity and Na^+^/K^+^ pump activity and reduced mitochondrial integrity [2].

ROS accumulation due to strenuous exercise changes sarcohormesis by depleting reduced glutathione (GSH), resulting in an increase in exercise-induced PGC-1α transcription factors [47]. PGC1a overexpression leads to a significant increase in oxidative type muscle fibres and increases many enzymes involved in oxidative metabolism [48]. Although animal studies have shown that the depletion of the endogenous glutathione content leads to a disruption of the skeletal muscle redox defence, thereby triggering the PGC-1a gene expression and regulating the mitochondrial adaptation [49], further studies are needed to investigate these findings in human trials.

Muscle-specific microRNAs (myomiRs) are also involved in the regulation of sarcohormesis. MyomiRs are multifunctional, small RNAs classified as part of non-coding RNA and are vital for various cellular biological processes and modulation of gene expression, particularly post-transcriptional genes through negative inhibition and thus regulating the redox adaptation processes [50,51]. The involvement of myomiRs in exercise-dependent adaptations and ROS-mediated redox metabolism has been well documented [50]. For example, miyomiR-214 is well known to modulate redox metabolism by regulating several oxidants during muscle hypertrophy and weakening superoxide production via nicotinamide adenine dinucleotide phosphate (NADPH) oxidase (NOX)4 [52]. NOX enzymes, mainly NOX2 and NOX4 isoforms expressed in muscles, are well-known for their primary function in insulin and muscle contraction-induced oxidant production [53]. However, it also has a substantial effect on modulating the redox state, gene expression, skeletal muscle metabolism, and insulin transport regulation in skeletal muscle. NOX inhibition weakens metabolic adaptation to exercise by reducing exercise mediated intracellular signalling and gene expression [54].

In regulating MyomiRs, PGC1-a expression changes depending on whether the exercise is acute or chronic. For instance, an acute resistance exercise promotes several signalling pathways related to protein synthesis by downregulating myomiR-1, myomiR-133a, myomiR133b, and myomiR-206 expression, resulting in increased muscle protein synthesis [55]. Rather, endurance training causes upregulation of these myomiRs rapidly, in particular, miR-1 and miR-29b [56]. Studies have shown that myomiR-378 has a role in controlling mitochondrial metabolism by regulating PGC1-B during endurance exercise, which suggests a shift in both mitochondria (e.g., increased mitochondria count) and skeletal muscle (e.g., the fiber-type shift towards to type-2), [57] resulting in metabolic adaptations. However, its effect on skeletal muscle mass is relatively small. Studies on myomiRs and adaptive response are promising [51,57], but more studies are needed to reveal the modulatory effect of myomiRs on exercise-mediated adaptive metabolism.

The maintenance of sarcohormesis is vital for physiological, metabolic and cellular function and survival under extreme oxidative stress [3]. Therefore, it is essential to improve the skeletal adaptive response using endogenous adaptations by upregulating cytoprotective signal transduction cascades [58], or exogenous supplements with antioxidants such as vitamin C, E, [46] polyphenols including flavanols, and anthocyanidins [59], or a diet rich in antioxidants (e.g., a diet rich in vegetables, fruits, chocolate, nuts, and their products) [60,61].

## 4. The Possible Role of N-Acetylcysteine on Oxidative Stress and Redox Regulation

N-acetylcysteine (NAC) is a well-known glutathione precursor that has been classified by the Food and Drug Administration (FDA) as an antidote in the treatment of poisoning and as adjuvant therapy for bronchopulmonary disorders [62]. NAC has been used in various preclinical and clinical studies (Table 1). In recent years, the popularity of NAC has also increased among athletes due to its effective antioxidant properties [15]. NAC contributes to antioxidant defence by directly removing a limited amount of ROS with its sulfhydryl group (-SH), or indirectly by activating the regeneration of glutathione, stimulating the production of hydropersulfides, or regulating of cytokine synthesis by inhibiting nuclear factor kappa B (NF-KB) [63].

Various antioxidant supplements such as vitamin E, C and α-lipoic acid reduce exercise-induced ROS and muscle fatigue and act as a reactive oxygen scavenger, improving recovery and anti-inflammatory response [64]; however, they cannot provide cysteine for glutathione replenishment, an important endogenous antioxidant defence. At this point, NAC supplementation provides the solution as a cysteine derivative [17]. NAC supplementation is often preferred over supplementing glutathione itself because of its better bioavailability [63].

GSH is of great importance for redox homeostasis, due to its direct role as an oxidant scavenger and as a catalyst for several detoxifying cellular enzymes, such as glutathione transferase, glyoxalases 1 and 2, peroxidases, and glutathione S-transferases, serving as a precursor in regulating antioxidant defence in muscle and other tissues [65]. GSH synthesis is modulated by a negative feedback mechanism [66]; however, the production rate lag behind during conditions of elevated oxidants [65]. Strenuous, prolonged exercise causes a dramatic increase in oxidized glutathione concentrations by 34 to 320% compared to baseline concentrations [63]. NAC supplementation can increase the GSH/oxidized glutathione (GSSG) ratio by providing a cysteine donor for GSH synthesis [17].

Although numerous NAC studies explain the beneficial effects of NAC on the body antioxidant defence [67,68,69,70,71,72,73,74], the exact mechanism remains unclear (Figure 1). Several studies have also suggested that although NAC has not restored the GSH content of the cells, it still provides cryoprotection [63,68,69,70,71]. For instance, Mihm et al. [75] has revealed that NF-KB activation may be associated with cellular glutathione concentrations. At this point, NAC supplementation can modulate redox homeostasis by reducing the NF-KB and MAPK-mediated proinflammatory cytokine response during inflammation [76]. Thus, it provides the preservation of GSH availability.

**Table 1 antioxidants-10-00153-t001:** Effects of N-acetylcysteine on mitohormesis and sarcohormesis in athletic population.

Participants	Study Design	Dose/Application Type (Oral/Injection)	Performance Test	Measurements	Study Outcomes	Ref.
A healthy untrained man	A case report	NAC (oral) (1 × 75 mg/kg), for five days or no supplementation	a submaximal prolonged ergometer cycling trial until volitional exhaustion	Untargeted metabolite profiling by CE-ESI-MS, including GSSG, GSH, 3-MH, L-carnitine, acetyl-L-carnitine, and creatine	NAC 3-MH ↓ GSH Oxidative stress attenuation in erythrocytes	[68]
Seven healthy untrained subjects (6 men, 1 woman)	A double-blind, randomized, repeated-measures crossover design	NAC (an initial loading dose of 62.5 mg/kg/h for the first 15 min, followed by a constant infusion of 25 mg/kg/h for the next 80 min) infusion or plasebo	one hour cycling exercise sessions 7–14 days apart, (55 min at 65% VO_2_ peak plus 5 min at 85% VO_2_ peak)	Insulin in plasma (3 h later exercise by a 2-h hyper insulinemic euglycemic clamp (40 mIU/min/m^2^)) IL-6, NAC in plasma TGSH, GSSG, glycogen, PC, antibodies in muscle	after NAC supplementation ROS/PC ratio ↓ (13%) GSH/GSSG ratio ↑ (×2) Insulin sensitivity ↓ (5.9%) AMPK mediated AS160 phosphorylation p70S6K phosphorylation ↓ (48%)	[80]
Eight healthy untrained men	A counterbalance, double-blind, crossover design	NAC (125 mg/kg/h for 15 min, then 25 mg/kg/h for 20 min prior to and throughout exercise) infusion or placebo	An intermittent sprint test comprised four exercise bouts at 130% VO_2_ peak on the ergometer	GSH, hematologic parameters and electrolytes in plasma	NAC vs. placebo K^+^-to-work ratio ↑ NAC was lessened GSSG ↑ and GSH ↓ in plasma following exercise	[69]
Nine healthy untrained men	A before and after study design	NAC (4 × 200 mg/day) (oral), for two days and an additional 800 mg NAC on the test morning	Cycling exercises for 30 min at aerobic and anaerobic thresholds Maximal bicycle ergometer exercise	TGSH and GSSG levels in blood TBARS and net PSC in plasma Exercise-associated damage in leukocyte DNA	Exercise at maximal, anaerobic and aerobic thresholds GSSG ↑, PSC NAC (after maximal exercise) TBARS-, GSH-s pre-exercise PSC ↑	[70]
Twenty-nine healthy untrained men	A randomized controlled study design	NAC (2 × 600 mg) (oral) or placebo, after meals for 7 days	A graded exercise treadmill test before and after supplementation	Fatigue index TAC, lactate and TNF-a in plasma CK in serum	NAC vs. placebo Fatigue index at day 8 ↑ (81.42 ± 2.99% at day 0 to 90.67 ± 2.07% at day 8) Lactate response ↓ TAC (vs. TAC ↓ in the control group) CK ↑ and TNF-a ↑ (at both groups)	[72]
Ten healthy untrained men	A cross-over design	NAC (150 mg/kg/h) or plasebo infusion before experimental protocols	MVC of ankle dorsiflexors and during electrically evoked contractions of the tibialis anterior muscle Repetitive tetanic stimulations at 10 Hz (protocol 1) or 40 Hz (protocol 2) to create fatigue	10, 20, 30, 60, 90, 120, 180, 240, and 300 s after fatiguing contractions at muscle Protocol 1: At each time point, three 850-ms trains of 10 Hz stimuli were delivered at 1 s intervals Protocol 2: At each time point, three 650-ms trains of 40 Hz stimuli were delivered at 1 s intervals	NAC vs. placebo During low-frequency electrical stimulation: Fatigue inhibition in tibialis anterior During high frequency stimulation Muscle fatigue Acute recovery from fatigue Strength and contractile properties of unfatigued muscle	[81]
Fourteen healthy untrained men	A double-blind, placebo-controlled study design	NAC (12.5 mg/kg) (oral) plus vitamin C (10 mg/kg) or placebo for 7 days immediately following muscle injury	An eccentric arm muscle injury to trigger acute-phase inflammatory response	Bleomycin detectable iron in serum SOD, GPX, 8-Iso-PGF2a, TAC, MPO, IL-6, CK and LDH in plasma	NAC+ vit C vs. placebo Bleomycin detectable iron ↑ LDH ↑ CK ↑ 8-Iso-PGF2a ↑ (immediately after and two days after exercise)	[82]
Seventeen healthy untrained men	A double-blinded, placebo-controlled, crossover design	Study 1: NAC (2 × 300 mg or 2 × 600 mg (~9, or ~18 mg/kg BW) (oral) or placebo previous day before exercise Study 2: NAC (oral) (35, 70, or 140 mg/kg BW)or placebo	For both studies: A bout of fatiguing handgrip exercise (three MVC of ~5 s duration separated by 1 min rest followed by a sequence of repetitive isometric contractions (3 s on, 3 s off) until the subject failed to reach the target force (70% MVC)	GSH, GSSG, CyS, cystine and CySSG in plasma	Study 1: Cys GSH Study 2: NAC (both 70 mg/ kg and 140 mg/ kg); CyS/total CyS ratio ↑ Side effects (only 140 mg/kg)	[83]
Twenty-nine healthy untrained men	A single-blind, placebo-controlled study design	NAC (10 mg/kg BW) (oral) for 21 days or NAC plus placebo (14 d NAC + 7 d placebo) or plasebo, for 14 days before exercise and for 7 days post-exercise	Eccentric exercise (3 sets until exhaustion (elbow flexion and extension on the Scott bench, 80% 1RM)	Muscle soreness using a Visual Analog Scale MDA, TNF-a, IL-10, PC in serum	All groups; MDA ↑ and PC ↑ (on days 4 and 7) Muscle pain ↑ and TNF-a ↑ (on day 2) TNF-a ↓ (after day 2) IL-10 ↑ on day 4, but ↓ after day 4 NAC; Maintained higher IL-10 after day 4	[84]
Eighteen subjects, both trained or untrained, (8 women, 10 men)	A placebo-controlled study design	NAC (150 mg/kg) (oral) or placebo before exercise	a bout of fatiguing handgrip exercise (MVC using three 5-s efforts at 30-s intervals, until the subject failed to reach the target force (70% MVC) followed by a sequence of repetitive isometric maneuvers	Free reduced NAC and Cys in plasma GSH and GSSG concentrations in erythrocytes	NAC; NAC ↑, Cys ↑ delayed fatigue (130% from baseline GSSG ↓	[74]
Thirty-six recreationally trained young men categorized by glutathione levels (*n* = 12 low, *n* = 12 moderate, *n* = 12 high)	A double-blind cross-over design	NAC (oral) (2 × 600 mg/day) or plasebo, for 30 days, 60-d washout period	Three whole body performance test (VO_2_ max, Wingate test, time trial cycling for 45 min with 60 rpm at 70% Wmax and subsequently performing as much work as possible for 15 min)	F2-isoprostanes in urine PC in plasma GSH, GPX, GR, SOD, catalase and NADPH in erythrocytes	Low GSH group; GSH ↑ (36%), SOD ↑ (37%), F2-isoprostanes ↓ (22%), PC ↓ (18%), GPX ↑ (26%), catalase ↑ (21%), GR ↑ (37%), NADPH ↑ (23%) Moderate GSH group; F2-isoprostanes ↓ (14%) No change other biomarkers High GSH groups; No change	[85]
Twelve recreationally trained men	A pair-matched design	NAC (oral) (2 × 50 mg/kg BW) or placebo supplementation, for six days before exercise	3 testing sessions (a pre-exercise IKD test, a damaging intermittent-exercise protocol, YIRT-L1, and a post-exercise IKD test), on alternating days	CK in plasma	NAC; CK ↑	[86]
Eight recreationally trained men	a double-blind, cross-over design	NAC (125 mg/kg/h for 15 min before exercise, then 25 mg/kg/h throughout exercise) infusion or placebo	A series of "step" cycle exercise tests that comprised two bouts of moderate-intensity exercise and one bout of severe intensity exercise	TSH and NO_2−_ in plasma Blood lactate and plasma K^+^ concentrations Pulmonary O_2_ uptake	NAC; TSH ↑ (225%) and remained elevated plasma NO^2-^ VO_2_ kinetics	[87]
Seven recreationally trained men	A double-blind, crossover design	NAC (125 mg/kg/h for 15 min, then 25 mg/kg/h for 20 min prior to and throughout exercise) infusion or placebo	Prolonged, submaximal exercise (cycling for 45 min at 70% VO_2_ peak, then continued at 90% VO_2_ peak until exhaustion)	Total NAC concentration, Hb, hematocrit, and electrolytes in plasma	NAC; No alteration in hematology, acid–base status, or plasma electrolytes plasma K^+^ ↓, K^+^-to-work ratio (at fatigue)	[88]
Ten recreationally trained men	A double-blind, crossover design	NAC (20 mg/kg BW) (oral) after muscle-damaging exercise for eight days	A muscle-damaging exercise, comprised 300 eccentric contractions (20 sets, 15 repetitions/set, 30-s rest between sets) with the quadriceps muscle group at a speed of 308/s on an isokinetic dynamometer	GSH, GSSG, PC, and TBARS concentrations in muscle Monoclonal anti-phospho-AKT^(Ser473)^, mTOR^(Ser2448)^, p70S6K ^(Thr389)^, p38MAPK ^(Thr180-Tyr182)^, NF-kB ^(Ser536)^, and TNF-a in muscle Cytokines (IL-1b, IL-6, IL-8, and IL-10), CRP, and CK activity in serum	NAC; CK ↓, CRP ↓, NF-KB phosphorylation ↓ (at 2 d after exercise) mTOR phosphorylation ↑, PKB ↑, p70S6K ↑ and p38MAPK ↑ (at 2 and 8 d after exercise), MyoD ↓ (8 d after exercise) TNF-a ↓ (at 8 days after exercise)	[13]
Eighteen recreationally trained male college students	A placebo-controlled experimental design	NAC (70mg/kg BW) (oral), placebo, or no application (control group), 90 min prior to experimental protocol	A repeat sprint protocol consisted of 12 × 30 m sprints separated by 35 s of passive recovery	Fatigue index	No change	[89]
Nine Recreationally Trained healthy men	A randomized, double blind cross-over study design	NAC (1800 mg) (oral) or placebo 45 min prior to the exercise protocol	Two 30 min constant load (85% VO_2_ peak), discontinuous exercises, separated by a 7–14 day interval	GSH in plasma To determine respiratory muscle strength; PImax, measured from RV PEmax, measured from TLC	NAC vs. placebo; No difference in PImax (at rest) PImax ↑ (14%) at 25 and 30 min (during exercise)	[90]
Nine healthy recreationally trained men	A double-blind randomized cross-over design	Either NAC or saline (placebo) intravenous infusion at 125 mg/kg/h for 15 min, then 25 mg/kg/h for 20 min prior to and throughout exercise	a prolonged moderate-intensity exercise (an ergometer cycling for 80 min at 62 ± 1% peak oxygen consumption (VO_2_ peak))	Lactate, glucose, FFA, insulin, total thiols, and NAC concentrations in plasma Muscle glycogen, muscle metabolites, thiols (GSH, GSSG, Cys), NAC, AMPK, and ACC-B in muscle biopsy	NAC; NAC ↑ and Cys ↑ (in both plasma and muscle during exercise) Muscle GSSG-, GSH- The increase of S-glutathionylation of a protein band ~270 kDa ↓ Glucose disposal −, lactate −, FFA −, insulin − No effect of NAC on the elevation in muscle AMPK and ACC-B levels during exercise (~3- and ~6-fold, respectively)	[91]
Thirty healthy, active, non-resistance trained men	A double blind, plasebo controlled, parallel study design	NAC (1800 mg/day) (oral), EGCG (1800 mg/day); or placebo throughout the 14-day supplementation period prior to exercise	One eccentric exercise bout (100 repetitions at 30°/s) using the dominant knee extensors	Perceived soreness (assessed with 10 cm scale) LDH and CK, SOD, serum cortisol, neutrophil counts, the neutrophil: lymphocyte ratio and TNF-a in serum Markers of intramuscular mitochondrial and cytosolic apoptosis including bax, bcl-2, cytochrome C, caspase-3 content enzyme activity, and total DNA content in muscle cell lysates	All time points Muscle soreness ↑ Less muscle soreness in the EGCG and NAC groups at 24 h after exercise Peak torque production ↓ at 6 and 24 h after exercise LDH ↑ at 6 h after exercise CK ↑ at 6, 24, and 48 h after exercise Muscle bax and bcl-2 muscle levels ↑ Caspase-3 enzyme activity ↑ (at 48 h after exercise) NAC vs. EGCG; Neu: Lym ratio ↑ (48 h after exercise) No changes in cytochrome C, caspase-3 content, caspase-3 enzyme activity and total DNA	[92]
Twenty-eight healthy trained men	A randomized, double-blind, placebo-controlled, cross-over design	NAC (2 × 600 mg) (oral) or ALA (2 × 300 mg) or placebo (2 × 350 mg) for eight days	Training sessions during eight days of supplementation (training content not specified)	PC and TAC, TBARS, EPO levels in plasma Blood GSH concentration Hb, Hct, erythrocytes, MCV, and MCH in complete blood count	NAC and ALA vs. placebo; PC ↓ and TBARS ↓ NAC vs. ALA and placebo; TAC ↑ 4 times more than the ALA group MCV ↑ and MCH ↑ (more than 12%) NAC vs. placebo; GSH ↑ (+33%), EPO ↑ (+26%), Hb ↑ (+9%) and Hct ↑ (+9%)	[71]
Ten healthy trained men	A two-trial, double-blind, crossover, repeated measures design	NAC (20 mg/kg/day) (oral) in three dosages (Immediately after exercise and 8 days thereafter)	A muscle-damaging exercise protocol (300 eccentric unilateral repetitions (performed in 20 sets of 15 repetitions/set with a 30 s rest interval between sets) of knee extensors at a velocity of 30°/s on an isokinetic dynamometer)	PC, sVCAM-1 in plasma GSH, GSSG, TBARS, CAT in erythrocytes cytokines (IL-1B and IL-6), CRP, TAC, and CK in serum Macrophages in blood	NAC; PC ↓, erythrocyte TBARS ↓, GSSG ↓, and serum TAC ↓ neutrophil and leukocyte count ↓ HLA+ and CD11B+ macrophages ↓ Upregulation of B lymphocytes ↓	[93]
Eight endurance-trained healthy men	A double-blind, randomized, crossover design	NAC (125 mg/kg/h for 15 min, then 25 mg/kg/h for 20 min prior to exercise) infusion or placebo	Two submaximal tests, separated by a 7-day interval, comprising cycling at 71% VO_2_ peak for 45 min	Muscle thiols, maximal Na^+^-K^+^ pump activity and Na^+^-K^+^ pump isoform mRNA including α1, α2, α3, β1, β2, and β3 mRNA	α3 ↓, β1 ↓, and β2 mRNA ↓ (2.0 to 3.4 times) by exercise, independent of NAC NAC vs. placebo; α2 mRNA ↓ (0.40-fold) α1 or β3 mRNA - The increase in Na^+^-K^+^ pump α2 mRNA with exercise in human vastus lateralis muscle ↓	[94]
Eight endurance-trained men	A double-blind, randomized, crossover design	NAC (125 mg/kg/h for 15 min, then 25 mg/kg/h for 20 min prior to and throughout exercise) infusion or placebo	a submaximal ergometer cycling for 45 min at 71% VO_2_max, then continued at 92% until exhaustion	K^+^ and other electrolytes in plasma Na^+^, K^+^-pump activity (K^+^-stimulated 3-OMFP in muscle cell lysates	Rise in plasma K^+^ during exercise and the ΔK^+^-to-work ratio at fatigue were decreased by NAC treatment	[95]
Eight endurance-trained men	A double-blind, randomized, crossover design	NAC (125 mg/kg/h for 15 min, then 25 mg/kg/h for 20 min prior to and throughout exercise) infusion or placebo	A submaximal ergometer cycling for 45 min at 71% VO2max, then continued at 92% until exhaustion	Reduced and total thiols, including NAC, TGSH, GSH, and Cys in blood and plasma Thiols, including NAC, GSH, and Cys in muscle cell lysates	NAC vs. placebo; Muscle total and reduced NAC ↑ (at fatigue and 45 min after exercise) Muscle Cys and cystine ↑ Blood GSH concentrations- Decrease in muscle TGSH ↓	[17]
Eight endurance-trained men	A double-blind, randomized, crossover design	Either NAC or placebo (saline) intravenous infusion at 125 mg/kg/h for 15 min, then 25 mg/kg/h for 20 min prior to and throughout exercise	a submaximal ergometer cycling for 45 min at 71% VO2max, then continued at 92% until exhaustion	JNK^(Thr183/Tyr185)^, ERK1/2^(Thr202/Tyr204) (Thr185/Tyr187)^, p38 MAPK^(Thr180/Tyr182)^ and NF-KB p65^(Ser536)^ in muscle cell lysates mRNA expression of MnSOD, IL-6, MCP, HSP70, PGC1-a	NAC vs. placebo; JNK phosphorylation ↓ Suppressed mRNA expression of MnSOD, but not ERK1/2, or p38 MAPK after exercise NF-KB p65 phosphorylation ↓ (14%) at fatigue	[96]
Seventeen semi-elite male rugby players	A double-blind, pre-post-controlled trial design	NAC (1 g/day) (oral) or placebo for six days	A broken bronco exercise test (a kind of muscle-damaging exercise) on days 5 and 6 of supplementation	A modified Muscle Pain and Treatment Satisfaction Questionnaire	NAC vs. placebo; muscle soreness ↓ (after first damaging exercise) Muscle pain ↑ (after second damaging exercise)	[97]
Twenty male volleyball athletes	A placebo-controlled study design	NAC (2 × 600 mg) (oral) or plasebo, twice daily, for seven days	A physical training session (consisted of 60 min of training (comprising flexibility exercise, technical part, and coordination work with play in short games), followed by 30 min of continuous running at relative intensity between an aerobic threshold of 1 and 2, and lastly strength endurance work for upper and lower limbs (5 sets of 20 repetitions at 60% of 1RM for upper limbs, and 6 sets of 20 repetitions at 65% of 1RM for the lower limbs))	GPX, SOD, GSH, TGSH in erythrocyte FRAP assay, LOOH, CK, AST, ALT, LDH, and ALP in serum TBARS, PC in plasma	NAC vs. placebo; post-exercise GPX and SOD ↓ TGSH ↑, GSH ↑, FRAP ↑ PC ↓	[98]
Ten well-trained triathletes	A double-blind randomized placebo-controlled crossover design	NAC (2 × 600 mg/day) (oral) or placebo for 9 days	105-min fatigue-inducing cycle protocol pre-and post-supplementation intensive training during the supplementation period	TAC, FRAP, GSH-to-GSSG ratio, TBARS, XO, hypoxanthine, IL-6, and MCP-1 in plasma NF-KB in mononuclear cell extracts F2t-isoprostane in urine	NAC vs. placebo; TAC ↑, TBARS ↓, urinary F2t isoprostanes ↓, IL-6 ↓ and MCP-1 ↓ post-exercise NF-KB activation ↑	[73]
Nineteen trained male oarsmen	A double-blinded randomized plasebo controlled design	NAC (6 g/day for three days) (oral) or placebo before the exercise	Six-minute maximum ergometer row set to simulate 2000 m water race	Concentration of neutrophils, plasma Cys and GSH in peripheral venous sample PaCO_2_, PaO_2_, pH, Hb, O_2_ saturation, and the concentrations of Hb and lactate in arterial blood sample	NAC vs. placebo; Cys ↑ neutrophil oxidative burst ↓ (±7 ± 6% vs. 17 ± 8%) No changes at work capacity, pulmonary gas exchange, or post-exercise reduction in pulmonary diffusing capacity	[99]
Eleven well-trained cyclists	A double-blinded crossover design	NAC (20 mg/kg BW) or placebo before exercise	repeated intense endurance cycling performance (4-min max test (PT-1), followed 90 min later by the second performance test (PT-2) in the form of the 4-min time trial)	lactate and K^+^ concentrations in whole blood count TAC in plasma	No change in plasma TAC and K^+^ levels	[100]
Nine well-trained male cyclists	A double-blind, repeated-measures, randomized crossover trial	NAC (5 × 100 mg/kg) or placebo drink; two doses on each of the two days before exercise, and last dose 30 min before exercise	6 × 5 min High-intensity-interval exercise ((HIIE) bouts at 82% PPO (316 ± 40 W) separated by 1 min at 100 W, and then after 2 min of recovery at 100 W, self-paced 10-min time-trial)	NAC, GSH, GSSG, TBARS, FFA in plasma Glucose, lactate in blood	NAC vs. placebo; GSH - TBARS ↓ Hb ↑, blood glucose ↑, plasma FFA ↓, blood lactate ↓	[101]

NAC: N-acetylcysteine; PC: Protein carbonyls; GSH: reduced glutathione; GPX: glutathione peroxidase; GR: glutathione reductase; SOD: superoxide dismutase; NADPH: Nicotinamide adenine dinucleotide phosphate; CR: Creatine kinase; IKD: isokinetic dynamometry; YIRT-L1: Yo-Yo Intermittent Recovery Test Level 1; CE-ESI-MS: capillary electrophoresis-electrospray ionization-mass spectrometry; GSSG: oxidized glutathione; 3MH: 3-methylhistidine; IL-6: Interleukine-6; TGSH: total glutathione; AS160: Akt substrate of 160 kDA; 3-OMFP: 3-o-methyl fluorescein phosphatase; MCP: Monocyte chemotactic protein; HSP-70: Heat-shock protein 70; PGC-1a: Peroxisome proliferator-activated receptor coactivator-1a, TSH: total sulfhydryl groups; Hb: hemoglobin; Hct: hematocrit, Cys: cysteine, PaCO2: CO2 partial pressure; PaO2:O2 partial pressure; TBARS: Thiobarbituric acid-reactive substances; PSC: peroxyl radical scavenging capacity; mTOR: polyclonal anti-phospho- mammalian target of rapamycin; p38 MAPK: mitogen activated protein kinase; NF-kB: Nuclear factor kappa B; TNF-a: tumor necrosis factor-a; CRP: C-reactive protein; CK: creatine kinase; TAC: total antioxidant capacity; EPO: erythropoietin; MCV: mean corpuscular volume; MCH: mean corpuscular hemoglobin; ALA: alpha-lipoic acid; MVC: Maximum voluntary contraction; 8-Iso-PGF2a: 8-iso prostaglandin F2a; SOD: Superoxide dismutase; LDH: lactate dehydrogenase; MPO: myeloperoxidase; FRAP: ferric reducing ability of plasma; XO: xanthine oxidase; sVCAM-1:Circulating Vascular Cell Adhesion Molecule-1;CAT:catalase; HLA: human leukocyte antigen; CD11b:(integrin αM); FFA: free-fatty acid; AMPK:AMP activated protein kinase; ACCB: Acetyl CoA Carboxylase beta; CySSG: cysteine-glutathione disulfide; BW: body weight; 1RM: 1 repeated maximum; MDA: malondialdehyde; PImax: Maximal inspiratory pressure; RV residual volume; PEmax: maximal expiratory pressure; TLC: total lung capacity; EGCG: epigallocatechin gallate; LOOH: lipid hydroperoxides; AST: aspartate aminotransferase; ALT: alanine aminotransferase; ALP: alkaline phosphatase; p70S6K: 70 kDa ribosomal protein S6 kinase, PKB: protein kinase B.Ref: references.

Multiple reversible posttranslational modifications occur in cells during redox adaptive processes including S-nitrosylation [77], covalent attachment of an NO group to a reactive cysteine thiol, S-Glutathionylation [78], post-translational binding of a glutathione tripeptide to a protein cysteine residue, and disulfide formation. S-nitrosylation and S-glutathionylation play an extensive role in regulating molecular signalling involved in muscle contraction and redox adaptation in skeletal muscle [21,77,78]. Additionally, an increase in intracellular NAC triggers the desulfurization of NAC-derived cysteine, leading to the production of hydropersulfides, which are then oxidized to sulfane sulfur species in the mitochondria [79]. These sulfane sulfur species are referred to as orchestra chefs, which modulate the antioxidant and cytoprotective effects of NAC in cells [79].

### 4.1. Possible Role of N-Acetylcysteine During the Regulation of Oxidative Stress Response of Mitochondria to Exercise

It is well documented that mitohormesis may not be sustainable under conditions where the mitochondria are exposed to excessive ROS produced in response to exercise [6]. At this point, the use of exogenous antioxidants begins to be considered. NAC supplementation has extensively studied the effects of NAC on body antioxidant defence in different populations using a variety of doses, periods, duration, and exercise protocols [67]. In this section, we will discuss NAC studies addressing the effects of NAC supplementation on the endogenous antioxidant system and adaptive response.

NAC supplementation was administered to several specific groups of athletes, including cyclists [100,101], triathletes [73], rowers [99], and volleyball players [98] to determine their role in redox related changes. Researchers also evaluated the possible effects of NAC on exercise-induced oxidative stress and adaptive mechanisms in healthy untrained men [68,69,70,72,74,80,81,82,83,84], recreationally trained men [13,74,85,86,87,88,89,90,91,92], and endurance-trained men [17,94,95,96]. Most studies primarily focused on the effects of NAC supplementation on GSH and GSH-related biomarkers in muscle [13,80,91,94] and plasma [68,69,70,74,82,83,85,90,91,98,99,101], and total or reduced NAC in muscle [17,101] and plasma [17,74,88]. However, it is challenging to comment on the relationship between NAC and glutathione availability, as studies differ in dose [17,83,93,98,99], exercise type [68,69,83,91], and training status [83,91,98,101]. It allows us to compare the results of the study in the case of studies with similar populations with similar NAC dosage and exercise protocol, thus shedding light on our interpretation. For instance; one study investigating the impact of NAC on fatiguing handgrip exercise in healthy untrained men in a dose-dependent manner reported that while low-dose oral NAC (9 or 18 mg/kg/day) previous day prior to exercise did not affect plasma Cys and GSH levels, high-dose oral NAC (70 or 140 mg/kg/day) increased the Cys/total Cys ratio [83]. In another study planned in a similar design and exercise protocol, a higher NAC dose (150 mg/kg) led to an increase in plasma NAC, Cys levels, and attenuation in GSSG concentration [74]. However, it should be noted that although a higher dose of NAC before fatigue exercise appears to be more effective, its use at a dose higher than 70 mg/kg also triggers gastrointestinal side effects [83]. Additionally, one case report applying a high dose of NAC (75 mg/kg) before a prolonged submaximal exercise reported no impact on GSH concentrations [68]. However, although it did not alter GSH status, it attenuated 3-methyl histidine (3-MH), and oxidative stress in erythrocytes. Since we know that NAC supplementation does not only offer its beneficial effects by raising GSH status [102], we can interpret the case report as NAC is useful in suppressing oxidative stress after a prolonged submaximal exercise due to its other beneficial impacts on redox status.

Importantly, one study highlighted that body glutathione levels prior to supplementation also had a substantial influence in evaluating the possible effects of NAC after whole-body exercise [85]. Researchers categorized participants according to current glutathione status in erythrocytes as low, moderate and high. Findings revealed that GSH (36%), SOD (37%) GPX (26%), GR (37%), and NADPH (23%) concentrations were increased in the low glutathione group, whereas only a change observed in the moderate GSH group was a decrease in F2-isoprostanes by 14% and no change was recorded in the high GSH group [85]. Thus, it indicates that participants with low GSH get more advantages from NAC supplementation. Additionally, one study also reported that NAC supplementation caused deleterious effects by lowering the GSH/GSSG-ratio [69]. Collectively, although most studies generally resulted in the beneficial effects of NAC on plasma redox parameters, some of them highlighted its detrimental roles on metabolic adaptations [13,80]. Therefore, we need more clarification on the potential effects of NAC in a whole-metabolic perspective.

Along with the glutathione status, several oxidant- and antioxidant biomarkers were measured to investigate the NAC effects on redox status by administering NAC before, during or after a study exercise protocol [70,93,98]. A study evaluating the effects of NAC pre-supplementation (4 × 200 mg/day for two days and one last dose of 800 mg on test morning) on maximal bicycle exercise in healthy untrained men showed that, although NAC pre-supplementation did not affect blood GSH levels, it led to an increase in peroxyl radical scavenging capacity (PSC) before the exercise, indicating an increase in the endogenous antioxidant capacity regardless of GSH concentrations [70]. Another study on male volleyball athletes revealed that pre-supplementation with NAC (2 × 600 mg/day) for seven days before a physical training session led a significant decrease in protein carbonyl levels [98]. Although post-exercise GPx and SOD levels before NAC supplementation was higher than post-exercise levels after supplementation, total glutathione, reduced glutathione, and ferric reducing ability of plasma (FRAP) level increased after NAC supplementation [98]. These studies indicated that NAC pre-supplementation generally acts to increase endogenous antioxidant capacity, either directly by fighting elevated oxidants and attenuating proinflammatory cytokines, or indirectly by increasing glutathione synthesis or by inhibiting the rise of exercise-induced oxidants. On the other side, NAC supplementation after exercise is also considered to be an effective strategy for modulating hormesis. One study aimed to investigate the possible influence of NAC supplementation immediately after and eight days thereafter a muscle-damaging exercise in healthy trained men [93]. Results revealed that NAC modulated the exercise-induced oxidative alterations by lowering the rise of plasma protein carbonyls (PC), erythrocyte thiobarbituric acid-reactive substances (TBARS), GSSG, and serum total antioxidant capacity (TAC). However, NAC also lowered the exercise-induced increase of total macrophages, including HLA+ and 11B+ macrophages in which are redox-sensitive innate immune macrophages, decreased the increase of neutrophil and leukocyte count, and disturbed the exercise-induced upregulation of B-lymphocytes. Thus, it suggested that NAC blunted the up-regulation of exercise-induced adaptive pathways. Therefore, although post-exercise NAC supplementation was administered to accelerate rapid recovery, its possible side effects on adaptive response should be kept in mind.

NAC supplementation may not alter biomarkers related to mitochondrial biogenesis after acute exercise [67,73]. One study related to metabolic adaptation showed that supplementing NAC at a dose of 1200 mg for nine days before a fatigue-inducing cycling exercise in well-trained triathletes (1) enhanced plasma total antioxidant capacity; (2) lowered pro-oxidant biomarkers determined by plasma TBARS and urinary F2t isoprostane levels; (3) attenuated inflammation measured by IL-6 and monocyte chemotactic protein 1; and (4) facilitated post-exercise NF-KB activation, thereby up-regulating exercise-induced redox alterations and adaptive process [73]. Collectively, studies on NAC and adaptive response remain controversial and require further investigation.

Combining NAC-supplements with other antioxidants is another interesting issue in regulating body antioxidant defences. Childs et al. [82] evaluated the supplementation with NAC (12.5 mg/kg) plus vitamin C (Vit C; 10 mg/kg) on an acute muscle injury induced by eccentric exercise in healthy untrained men. Results showed that concentrations of bleomycin detectable iron in serum and lactate dehydrogenase (LDH) and creatine kinase (CK) in plasma increased more in the NAC plus vitamin C supplementation group than in the placebo group. The NAC plus vitamin C group had higher lipid hydroperoxides and 8-Iso-PGF2a levels in plasma two days after exercise. These findings suggest that vitamin C and NAC supplementation triggers exercise-induced oxidative stress and muscle cell damage. This may be due to the fact that excessive intake of exogenous antioxidants may serve as a toxic agent and may even trigger rather than prevent oxidative damage—termed the “antioxidant paradox”. This term implies that although the consumption of dietary antioxidants may provide beneficial effects in combating ROS to alleviate oxidative damage, high-dose antioxidant supplements can have detrimental consequences. One example is that oxidative radical damage may develop in both those who consume less than the recommended amount of vitamin C and those who take excessive vitamin C supplements [103].

### 4.2. Possible Part of N-Acetylcysteine During the Regulation of Oxidative Stress Response of Skeletal Muscle to Exercise

In addition to mitochondria-mediated adaptive changes, many factors such as cytoprotective gene adaptations, mRNA-mediated modulations, insulin sensitivity and glucose uptake into skeletal muscle, K+ regulation, muscle pain and fatigue play unique roles in sarcohormesis against excessive oxidative stress [3,51,104]. In this Section, we briefly discuss several factors involved in the regulation of sarcohormesis.

Extreme conditions such as strenuous exercise, eccentric exercise, or injury stimulate intracellular signalling pathways that rapidly alter the redox state by activating NOXs, which causes ROS to increase, and thus myofibers are injured [28]. However, at the same time, anti-inflammatory cytokine performed by neutrophils and endogenous anti-inflammatory enzymes (e.g., xanthine oxidase, glutathione peroxidase and cyclooxygenase-2) activated by injured myofibers generate an anti-inflammatory response to modulate the redox state [41]. The maintenance of sarcohormesis depends on the adaptive ability and availability of cytoprotective signalling factors within the muscles [5]. Situations that endogenous response is not sufficient to maintain sarcohormesis, NAC supplementation may assist these processes. Michailidis et al. [13] investigated the effects of oral NAC supplementation (20 mg/kg for 8 days after a muscle-damaging exercise) on human skeletal muscle signalling in recreationally trained men. Results presented that NAC supplementation decreased inflammatory biomarkers, including CK, C-reactive protein (CRP), and proinflammatory cytokines, and NF-KB phosphorylation at two days after exercise, and also decreased TNF-a at eight days after exercise. It also attenuated the increase in phosphorylation of mTOR, protein kinase B, p70 ribosomal S6 kinase, ribosomal protein S6, and p38MAPK at two and eight days after exercise. AKt/mTOR signalling cascade is of great importance in regenerating muscle damage after excessive prolonged exercise with Akt/MyoD-mediated muscle cell differentiation and regeneration. NAC supplementation along a eight-day recovery period after muscle-damaging exercise caused an impaired skeletal muscle inflammatory response by mainly attenuating Akt/mTOR phosphorylation, MyoD and p38MAPK phosphorylation and decreasing about 30% in neutrophil count and macrophage infiltration. Thus, indicating although NAC supplementation after muscle-damaging exercise elevated glutathione concentrations but it also blunted redox-dependent signalling pathways, and caused impaired skeletal muscle inflammatory response and capacity. One possible mechanism is that NAC supplementation alters redox-sensitive NF-KB signalling pathway activation by inhibiting exercise-induced mitogen-activated protein kinase p38 (p38 MAPK) [3], and phosphorylation of JNK [27]. This adaptive signaling agent orchestrates adaptive cellular responses to both intracellular and extracellular stresses in a ROS-dependent manner. In addition to that, one study found that NAC supplementation before and throughout a submaximal cycling exercise inhibited JNK phosphorylation. Therefore, supplementation of NAC after a muscle-damaging exercise or before and through submaximal exercise can be a null strategy that causes harm rather than benefits.

Post-translational redox modifications of cysteine thiols on various proteins, including S-glutathionylation [78], serve as a critical regulator of numerous redox adaptations and myogenic programming in skeletal muscle. As an important example, GSH plays a fundamental role in modulating the antioxidant defence of myosatellite cells under increased oxidative conditions by scavenging oxidative radicals and up-regulating survival mechanisms [63,105]. GSH is rapidly depleted under oxidative conditions, thus inducing a reduction of myogenesis by triggering NF-B activation that causes down-regulation of MyoD [106], a myogenic protein known for its unique role in promoting myoblast proliferation and differentiation. Increased intracellular GSH following NAC supplementation could be an effective strategy to increase skeletal muscle myogenesis and sarcohormesis by regulating MyoD via NF-KB activation.

MyomiRs-mediated modulation is considered as another key factor in maintaining sarcohormesis. Petersen et al. [96] determined the possible role of NAC infusion on early muscle adaptive response to a submaximal exercise in endurance athletes. Researcher injected NAC to the participants before and throughout the exercise. Findings revealed that an NAC infusion at a large dose suppressed the mRNA expression of MnSOD, but not ERK1/2, or p38 MAPK following exhaustive submaximal exercise, indicating the blunting effect of NAC on skeletal muscle cell gene expression and signalling cascades involved in early adaptations to exercise. Another study by the same research group with the same NAC dose protocol and same exercise type evaluated whether NAC-induced ROS scavenging blunted the elevation in Na^+^-K^+^-pump mRNA during submaximal exercise in human muscle [94]. Muscle lysates were used to analyze Na^+^-K^+^-pump α (1), α (2), α (3), β (1), β (2), and β (3) mRNA. Results presented that α2 mRNA decreased 0.40-fold in the NAC group. α3, β1, and β2 mRNA were attenuated 2.0 to 3.4 times by exercise, independent of the NAC infusion. Neither exercise nor NAC changed the α1 or β3 mRNA. NAC infusion reduced the increase in Na^+^-K^+^ pump α2 mRNA with exercise in the human vastus lateralis muscle, pointing out the perturbative effects of NAC on skeletal muscle gene expression. Thus, NAC supplementation does not appear to be suitable for myomiRs-mediated redox modulation in maintaining sarcohormesis.

The elevation of ROS as a result of a metabolically and physically demanding exercise leads to various changes in ion metabolism, including Ca^2+^ sensitivity and K^+^ dysregulation in contractile muscles and other tissues [94]. ROS causes oxidation of the Na^+^-K^+^-ATPase α/β subunits, resulting in the loss of K^+^, which impairs the membrane potential and the ability to contract. NAC is thought to be effective in modulating K^+^ homeostasis during ROS-mediated degradation [88]. However, NAC studies on K regulations presented equivocal results [69,88,95,100]. One study sought to evaluate the role of NAC on muscle Na^+^-K^+^-pump activity during submaximal cycling exercise in endurance-trained men [95]. According to the findings, a rise in plasma K^+^ during exercise and the Δ K^+^-to-work ratio at fatigue were decreased by NAC-treatment, indicating that NAC supplementation decreased muscle fatigue by improving K^+^-regulation, in accordance with a study by Medved et al. [88]. On the contrary, another study on the interaction between NAC supplementation and K^+^-regulation in plasma during exercise revealed that NAC led to an increase in the K^+^-to-work ratio, indicating that NAC impaired the K^+^-regulation during intense, intermittent exercise, thereby declining exercise performance [69]. In addition to that, one study on well-trained cyclists found that NAC supplementation before an repeated intense endurance cycling performance did not affect plasma K^+^-concentrations [100]. Collectively, further research is urgently needed to clarify the possible contributions of NAC on K^+^-regulation.

Recent literature discusses the effect of ROS on muscle glucose transport and insulin sensitivity [91,99,101]. Antioxidant supplementation can blunt acute exercise-mediated insulin sensitivity by reducing ROS [80], which is a paramount role for skeletal muscle to modulate the insulin response. NAC studies on insulin sensitivity and skeletal muscle glucose uptake have highlighted that NAC impairs muscle glucose uptake [101], decreases insulin sensitivity [80], or has no effect [91].

During ex vivo studies, contractile muscles are exposed to higher amounts of NAC, thereby inducing its regulatory role by modulating glucose uptake into skeletal muscle [107,108]; however, NAC bioavailability is reduced by approximately 90% in human studies compared to ex vivo [109]. Thus, one possible explanation for why NAC does not affect glucose uptake into the contractile muscle is that the increase in cysteine levels with NAC is not so effective at preventing the irregularity caused by ROS in glucose uptake.

Exercise duration and intensity are also of great importance in determining NAC effectiveness on glucose uptake. A study on glucose uptake and NAC supplementation highlighted that the reason why NAC was ineffective in regulating glucose uptake may be because the exercise protocol applied in the study (prolonged-moderate exercise) may not be sufficient to trigger excessive ROS accumulation that induces skeletal muscle redox signalling [91]. Therefore, more studies are needed to investigate the interaction between glucose uptake in skeletal muscle and NAC supplementation during recommended exercise protocol.

Elevated oxidants also cause further damage to the contractile muscles, causing a dramatic increase in muscle fatigue [37]. Several NAC studies have focused on its possible fatigue-eliminating mechanisms during muscle-damaging exercise (e.g., eccentric exercise) [82,84,92]. Acute eccentric exercise appears to significantly increase muscle damage, mitochondrial apoptosis markers, apoptotic enzyme activity, and whole blood cell inflammation markers with no change in oxidative stress [84]. In a study by Kerksick et al. [92] revealed that one eccentric exercise bout resulted in elevated muscle soreness. Researchers also evaluated the effects of pre-exercise NAC (1800 mg) and epigallocatechin gallate (EGCG) (1800 mg) supplementation on an eccentric exercise bout in active, non-resistance trained men. Although there was an increase in muscle soreness at all time points compared to baseline levels in all groups, less soreness levels were found in the EGCG and NAC groups than in the placebo group at 24 h after exercise. Another study determining the role of NAC for 14 days before and seven days after an eccentric exercise reported that a remarkable increase in MDA and carbonyl concentrations was reported on days four and seven after eccentric exercise regardless of NAC supplementation. On day 2 after eccentric exercise, muscle pain and a significant increase in TNF-a were observed in all groups and a decrease in the following days independent of NAC supplementation. Although IL-10 levels increased significantly on day 4 in all groups, only the NAC-supplemented groups maintained high IL-10 levels on day 7 after eccentric exercise. While these studies are promising that administration of NAC supplementation before and/or after eccentric exercise may reduce muscle fatigue, more research is needed to clarify the specific roles of NAC in muscle fatigue and oxidative stress response.

Training frequency is also crucial to the effectiveness of NAC. In a study by Rhodes et al. [97] evaluated the effectiveness of NAC supplementation on repeated bouts of high-intensity exercise. Results showed that six days of NAC supplementation relieved muscle soreness after a damaging exercise session; however, muscle soreness increased after the second damaging exercise. Additionally, a landmark study by Reid et al. [81] found that pre-exercise NAC supplementation acts as a selective inhibitor during exercise-induced muscle fatigue by inhibiting muscle fatigue in the tibialis anterior during repetitive, low-frequency electrical stimulation, but not effective in the recovery process after fatigue and during high-frequency electrical stimulation.

Although NAC supplementation appears to be beneficial for relieving muscle fatigue, one study has noted that NAC supplementation leads to increased CK levels in humans [86]. Researchers have claimed that one possible explanation for why plasma CK levels increase after NAC supplementation may be that NAC enables the muscle for more work, thus causing more muscle damage.

The reduction/or suppression of respiratory fatigue is also important in regulating the body hormetic balance [87,90,99]. Exercise-induced muscle damage can trigger respiratory failure, causing critical consequences on the respiratory muscles [90]. In this case, keeping peripheral fatigue as close to baseline levels as possible can be potentially advantageous for preventing an unwanted change in respiratory muscles during exercise. Therefore, antioxidant supplements are considered of interest as it is suggested to prevent the increase of peripheral fatigue [99]. In a study investigating the effects of NAC on respiratory muscle fatigue during heavy exercise claimed that there was no difference in terms of maximal inspiratory pressure (PImax) between the supplement and placebo group at rest [90]. During exercise, PImax was 14% lower in the placebo group compared to the supplement group at 25 and 30 min (*p* < 0.05), suggesting less respiratory muscle fatigue with NAC. NAC could be a beneficial agent for respiratory fatigue; however, it still needs further investigation.

### 4.3. Uncertainties on the Use of N-Acetylcysteine as an Antioxidant Supplement

Although considered safe to use as an antioxidant supplement [67], NAC’s effect on metabolism is still controversial. One of the main differences of NAC studies is that, unlike other antioxidants, it was administered to subjects usually by infusion [17,69,80,81,87,88,91,94,95,96]. According to the 2020 World Anti-Doping Agency Prohibited List [110], methods of intravenous infusion and/or injection of >100 mL during a 12-h period are prohibited except for those legally applied in relation to hospital treatments, surgical procedures or clinical diagnostic investigations. Therefore, the use of intravenous NAC in the athletic population is limited. Additionally, studies comparing the effect of an injection and oral supplementation of NAC on inflammation have revealed that although the injection has higher bioavailability, oral supplementation is highly recommended due to the side effects of the injection, including vomiting, headache, difficulty breathing, nausea, and heart and circulatory problems [67]. Oral NAC supplementation can also cause side effects such as nausea, vomiting, and diarrhoea, but these side effects are attenuated by interfering with dosage and timing. It has been observed that by dividing the daily dose of NAC into 2–4, the side effects of the supplement are reduced while creating a similar antioxidant response [67].

Another critical factor in NAC supplementation is its possible detrimental effect on redox-mediated adaptive signalling. Some studies hypothesize that NAC supplementation can support the body’s antioxidant defences without blunting the endogenous redox-mediated transcription pathways [63]. However, Michailidis et al. [13] rejected this hypothesis, revealing that NAC supplementation could inhibit the adaptive response in human skeletal muscle. Another study reported that while endurance exercise enhanced MnSOD gene expression, where an NAC infusion completely blocked the exercise-induced increase in mRNA expression of MnSOD in skeletal muscle [96]. To describe NAC supplementation as an effective antioxidant supplement for athletes, it should provide advantages for combating excessive ROS accumulation and at the same time not cause a blockage for exercise-induced nuclear and cytosolic adaptations. With this perspective, it still needs further investigation.

Dietary intake may also alter the effect of NAC on metabolism. For example, it is well documented that reduced thiols interact directly with nitric oxide [111], and S-Nitrosothiols are classified as a class of signalling molecules with several unique roles in intracellular signalling cascades, neurotransmission, and antimicrobial defence [112]. Therefore, combining NAC supplements with dietary nitrates may provide additional benefits by altering each other’s bioavailability. However, it should be kept in mind that these two components can blunt each other’s activities. Therefore, due to the lack of studies on this topic, we still cannot give a definitive recommendation on the beneficial effect of taking it together. Therefore, human studies are needed to elucidate the precise relationship between NAC and dietary intake.

Co-supplementation with NAC also matters for NAC metabolism. It is suggested that antioxidants rich in polyphenols may create a greater ergogenic effect on redox homeostasis if co-supplemented with NAC [82]. The possible mechanism is the effects of NAC on scavenging oxidants that rapidly degrades NO, or providing its sulfhydryl group to form an S-Nitrosothiols. However, there is limited research investigating these possible mechanisms in athletes. One study investigating the influence of multi-day pomegranate extract co-supplemented with NAC in male cyclists rejected the study hypothesis that it originally suggested, suggesting that co-supplementation may impair exercise performance by enhancing the formation of SNO, thus reducing NO bioavailability [113]. Further studies are needed to be elucidated the exact influence of possible interactions.

Additionally, it is known that low protein intake or hunger/restrictive eating leads to a decrease in glutathione intake and therefore, can cause a decrease in erythrocyte glutathione levels [114]. Most of NAC studies collected diet records to minimize confounding factors and standardize diet consumption and collect data at overnight fast in order to avoid the effect of diet on redox status. However, previous dietary glutathione intake should be monitored during NAC studies to account for its effect on blood glutathione levels.

The effects of NAC may alter according to the body’s current thiol status. Many antioxidant supplements are only effective on athletes who lack the antioxidant applied in their bodies. This claim has been proven by studies on vitamin C and glutathione [85,115]. For this reason, it is recommended that body thiol status should be measured before NAC supplementation.

Conflicts between NAC studies may also arise due to differences in the study protocol, including the type/dose/duration of the NAC supplement, training time/protocol, examining different samples (e.g., muscle lysates vs. blood), and training status. For instance, a study investigated exercise-dependent redox alterations in plasma and muscle lysates during three different exercise intensities including sprint-interval exercise (SIE), high-intensity interval exercise (HIIE), and continuous moderate-intensity exercise (CMIE). Findings revealed that biomarkers associated with plasma redox do not adequately reflect the skeletal muscle redox-sensitive protein signalling that regulates exercise-induced adaptations [116]. It should be considered when interpreting the study findings.

As it is known that several NAC studies have been done on animals [117,118], interpretation of the results of the studies to humans can also be difficult due to differences between species. Therefore, inter-species differences in the process of adaptation and genetic predisposition should be kept in mind during interpretation.

Applied NAC dose may change its efficiency on redox status and sports performance [83]. Multiple NAC doses are preferred in NAC studies. A new meta-analysis on NAC supplementation and sports performance reported that the dose of NAC supplements ranged from 1200 mg to 20 g/day in an experimental setting [67]. The researchers also stressed that the side effects of NAC supplementation remain unclear. Another study comparing dose-response interactions in terms of side effects, Ferreira et al. [83] found that a daily dose of 70 mg/kg NAC caused no side effects. As we know that high doses of NAC supplements can turn into pro-oxidants like other antioxidants, leading to impairment of adaptive cytoprotective responses, it is noteworthy that NAC dose should be carefully arranged to eliminate its side effects on the whole metabolism.

## 5. Conclusions

In this narrative review, we analyzed current literature in order to improve our understanding of the effects of NAC supplementation on mito- and sarcohormesis in-depth. Figure 1 summarizes the data from the studies presented in Table 1 showing the potential effects of NAC on mito- and sarcohormesis under exercise-induced oxidative stress. Exercise at different intensities, such as muscle-damaging [13,82,84,93,94,97], fatigue-inducing [17,68,73,74,80,83,88,90,95,96,100], high-intensity [87,101], and repetitive intermittent exercises [81,86,89], cause excessive accumulation of ROS in the body. NAC supplementation can provide several advantages for maintaining mitohormesis by increasing endogenous antioxidant defense, suppressing pro-oxidants, and upregulating cytoprotective adaptations [71,73,74,85,98,99]. It can also support sarcohormesis maintenance by upregulating cytoprotective adaptations, regulating myomiRs-mediated regulation, regulating insulin uptake/insulin sensitivity, reducing muscle fatigue/pain, and regulating plasma K^+^-levels [88,90,92,94]. In addition, NAC studies have conducted on recreationally trained individuals, endurance trained men, and athletes. Three studies investigating the effect of NAC on sarcohormesis in the athletic population did not confirm NAC-induced sarcohormesis in athletes [99,100,101]. Two of these studies argued that NAC supplementation had no effect on sarcohormesis [99,100], while one reported that it negatively affected ROS-induced metabolic adaptations in skeletal muscle [101]. In contrast, three studies investigating NAC-induced mitohormesis stated that NAC supplementation positively affected mitohormesis [73,98,99]. On the other hand, three out of four studies in endurance trained men supported NAC-induced sarcohormesis [17,94,95], while one claimed the opposite [96]. Additionally, it has been reported that NAC supplementation has no effect on mitohormesis in endurance trained men [17]. Considering all the findings, it appears that studies on NAC-induced mitohormesis and sarcohormesis are very few and suggest conflicting results, especially related to sarcohormesis. Furthermore, all studies in athletic population have been performed on men. Therefore, further studies need to evaluate the possible role of NAC in both mitochondria and muscles in redox regulation processes, and also perform on both gender.

Training adaptations occur by upregulation of endogenous antioxidant enzymes and heat shock proteins (HSP) in response to ROS accumulation [119]. However, in acute situations where prolonged training (low-frequency fatigue exercise) or competition lasts for a single or multiple-days (required rapid recovery), suppressing excess ROS accumulation may become the first priority to regulate metabolic and physiological mechanisms, including reduced muscle damage and fatigue, and thus accelerate muscle recovery. Taking all antioxidant supplementations into account, NAC may be the most effective and safe strategy to promote both mito-and sarcohormesis in athletes exposed to exercise-induced oxidative stress. However, it should be kept in mind that various practices such as NAC supplementation during training periods or long-term NAC supplementation can be a blinding factor for metabolic adaptations. Therefore, NAC application strategies should be planned in a way that does not prevent training-dependent metabolic adaptations.

Although the recommended NAC dose for athletes ranges from 1.2 to 5 g per day [67], there is no consensus on the optimum dose. Furthermore, as we mentioned earlier, exercise intensity and participant characteristics also differed between studies. With this in mind, more studies should be performed to determine the appropriate dose of NAC to be administered based on exercise intensity.

In addition, the response rate to NAC supplementation also depends on individual NAC bioavailability (e.g., absorption rate), previous thiol concentration, and the antioxidant defense system present in the body [63,85]. More research is needed to gain more insight into the dose-response relationship of NAC on body antioxidant status in the athletic population, taking into account the body’s current thiol and antioxidant availability.

It is well known that an advanced antioxidant defense system with metabolic adaptation against ROS is required for successful sports performance [9]. For this reason, the potential effects of antioxidant supplements have been extensively studied in order to achieve these goals and develop an effective antioxidant strategy [9,12,15]. However, the determination of the optimal exogenous dose for any antioxidant supplementation is rather complex, as the optimal ROS required to promote cytoprotective adaptation to exercise cannot be fully defined. Further, considering the antioxidant paradox and its negative consequences [103], excessive antioxidant supplements may result in detrimental alterations in sports performance and metabolic adaptation processes. Therefore, several studies have stated the substantial role of dietary antioxidants in maintaining physiological antioxidant status during intense training, suggesting that diets rich in antioxidants including vegetables, fruits, nuts, and whole grains and polyphenolic compounds may provide an effective opportunity to up-regulate the antioxidant defense system [9,120,121]. In certain situations such as multi-day races or prolonged extensive submaximal exercise, antioxidant supplementation may provide advantages. However; considering all the aforementioned data, the antioxidant content of the current diet should be examined before any supplement strategy is planned, and antioxidant supplements should be customized according to individual needs.

Although NAC supplements have gradually increased over the past decade and have the potential to improve redox status, their use is limited in practice. Since it is vital to maintain the hormetic balance against exercise to ensure the survival of mitochondria and myocytes, and NAC supplementation may seem to be a promising agent for modulating hormesis. However, the potential effects of NAC supplementation on cytoprotective signalling pathways and genes are still unclear and worth further investigation.

## Figures and Tables

**Figure 1 antioxidants-10-00153-f001:**
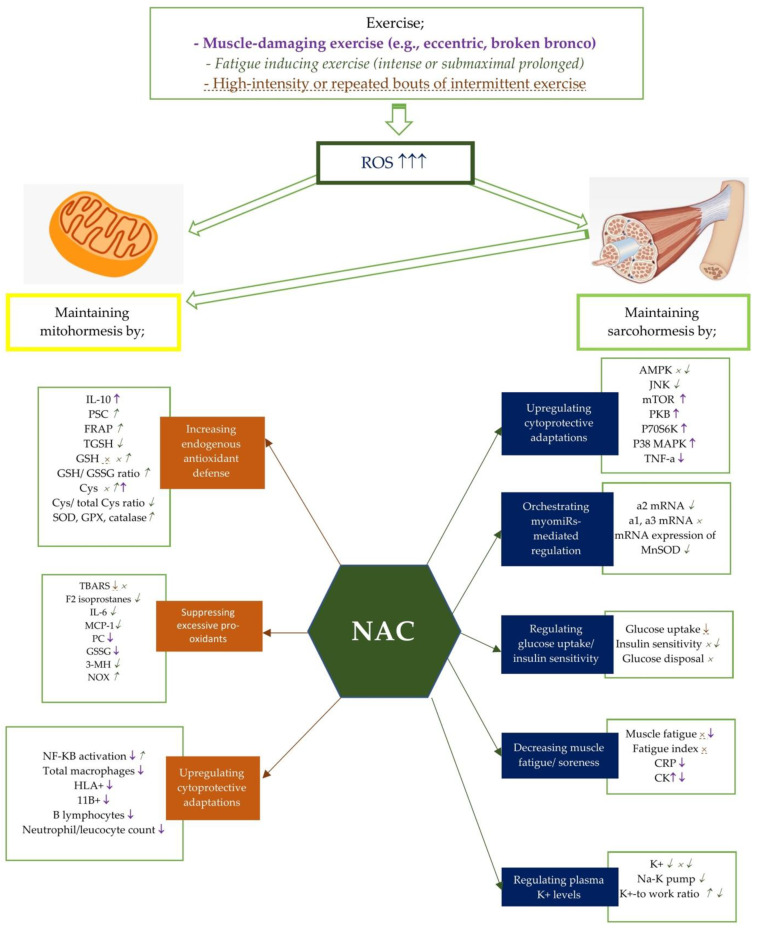
Potential effects of NAC on mito- and sarcohormesis under exercise-induced oxidative stress. The potential effects of NAC supplementation may vary according to the exercise severity. Therefore, NAC supplementation has been studied on certain exercise intensities including muscle-damaging exercise such as eccentric and broken bronco, fatigue inducing exercise, and high-intensity or repated bouts of intermittent exercise. Each exercise intensity is differently colored and shaped to represent changes in metabolic processes. ↑, ↓, and × represent the alterations caused by muscle damaging exercise (e.g., eccentric, broken bronco). 
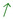
, 
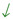
, and 
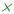
 represent the alterations caused by fatigue-inducing exercise (e.g., intense or submaximal prolonged). ↑, ↓, and × represent the alterations caused by high-intensity or repeated bouts of intermittent exercise. For example, NAC supplementation has showed an enhancing effect of NF-KB activity after fatigue-inducing exercise, while reducing plasma NF-KB activation after muscle-damaging exercise. Abbreviations: NAC: N-acetylcysteine; PC: Protein carbonyls; GSH: reduced glutathione; GPX: glutathione peroxidase; SOD: superoxide dismutase; NADPH: Nicotinamide adenine dinucleotide phosphate; CR: Creatine kinase; GSSG: oxidized glutathione; 3MH: 3-methylhistidine; IL-6: Interleukine-6; TGSH: total glutathione; MCP: Monocyte chemotactic protein; PGC-1a: Peroxisome proliferator-activated receptor coactivator-1a, Cys: cysteine, TBARS: Thiobarbituric acid-reactive substances; PSC: peroxyl radical scavenging capacity; mTOR: polyclonal anti-phospho- mammalian target of rapamycin; p38 MAPK: mitogen activated protein kinase; NF-kB: Nuclear factor kappa B; TNF-a: tumor necrosis factor-a; CRP: C-reactive protein; TAC: total antioxidant capacity; SOD: Superoxide dismutase; FRAP: ferric reducing ability of plasma; CAT: catalase; HLA: human leukocyte antigen; CD11b:(integrin αM); AMPK:AMP activated protein kinase; MDA: malondialdehyde; p70S6K: 70 kDa ribosomal protein S6 kinase, PKB: protein kinase B.

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
