# Peer review of "How N-Acetylcysteine Supplementation Affects Redox Regulation, Especially at Mitohormesis and Sarcohormesis Level: Current Perspective"

_antioxidants, 2021, doi:10.3390/antiox10020153_

Round 1

Reviewer 1 Report

I found this narrative review well written and structured. There are just few minor errors to be revised.

Below there are my comments:

  • Line 36. It is more correct to say that the endogenous response to exercise is dependent from the characteristics of exercise training program (i.e., type, intensity, frequency, duration).
  • LINE 94. Revise this sentense. Two times is repeated the word "fusion".
  • LINE 106. PLEASE REVISE THIS SENTENCE....."including improving".....
  • Line 168. This sentence is not clear. Please revise
  • Figure 1. Some part of the figure 1 is not readable. Moreover in the legend should be indicated the meaning of the symbols reported in each square. Please revise

Author Response

Antioxidants- Answers to Reviewer 1

Thank you for your kind comments and contribution. We have reorganized the paper throughout your suggestion. Thank you also for your generous introduction, we truly appreciate that you valued positively our work.

Below are our responses to your comments.

  • Line 36. It is more correct to say that the endogenous response to exercise is dependent from the characteristics of exercise training program (i.e., type, intensity, frequency, duration).
  • Answer: Line 36. According to your suggestion, we revised the sentence as stated below;

“Endogenous response to exercise depends on the characteristics of the exercise training program (i.e., type, intensity, frequency, duration).”

  • LINE 94. Revise this sentense. Two times is repeated the word "fusion"
  • Answer: LINE 94. We revised the sentence as stated below;

“It is dynamically regulated by processes of fission, fusion and mitophagy.”

  • LINE 106. PLEASE REVISE THIS SENTENCE....."including improving".....
  • Answer: LINE 106. We revised the sentence as stated below;

“Phosphorylated PGC-1a by redox-sensitive kinases modulates numerous muscle-related transcriptional factors through nuclear respiratory factors 1 and 2 (NRF-1 and NRF-2) [27], cAMP Response Element-Binding Protein (CREB) [28], Histone deacetylase 1 (HDAC1) [29], mitochondrial transcription factor A (mtTFA) [30], and myocyte enhancer factor-2 (MEF2) [31], by improving muscle metabolism and enhancing muscle performance.”

  • Line 168. This sentence is not clear. Please revise
  • Answer: Line 168. We revised the sentence as defined below;

“Muscle-specific microRNAs (myomiRs) are also involved in the regulation of sarcohormesis. MyomiRs are multifunctional, small RNAs classified as part of non-coding RNA and are vital for various cellular biological processes and modulation of gene expression, particularly post-transcriptional genes through negative inhibition and thus regulating the redox hormesis”.

  • Figure 1. Some part of the figure 1 is not readable. Moreover in the legend should be indicated the meaning of the symbols reported in each square. Please revise
  • Answer: We revised the figure and figure legends as stated below;

 Figure 1. Potential effects of NAC on mito- and sarcohormesis under exercise-induced oxidative stress. The potential effects of NAC supplementation may vary according to the exercise severity. Therefore, NAC supplementation has been studied on certain exercise intensities including muscle-damaging exercise such as eccentric and broken bronco, fatigue inducing exercise, and high-intensity or repeated bouts of intermittent exercise. Each exercise intensity is differently colored and shaped to represent changes in metabolic processes. ­, ¯, and ´ represent the alterations caused by muscle damaging exercise (e.g.; eccentric, broken bronco). ­, ¯, and ´ represent the alterations caused by fatigue-inducing exercise (e.g.; intense or submaximal prolonged). ­, ¯, and ´ represent the alterations caused by high-intensity or repeated bouts of intermittent exercise. For example, NAC supplementation has showed an enhancing effect of NF-KB activity after fatigue-inducing exercise, while reducing plasma NF-KB activation after muscle-damaging exercise. Abbreviations: NAC: N-acetylcysteine; PC: Protein carbonyls; GSH: reduced glutathione; GPX: glutathione peroxidase; SOD: superoxide dismutase; NADPH: Nicotinamide adenine dinucleotide phosphate; CR: Creatine kinase; GSSG: oxidized glutathione; 3MH: 3-methylhistidine; IL-6: Interleukine-6; TGSH: total glutathione; MCP: Monocyte chemotactic protein; PGC-1a: Peroxisome proliferator-activated receptor coactivator-1a, Cys: cysteine, TBARS: Thiobarbituric acid-reactive substances; PSC: peroxyl radical scavenging capacity; mTOR: polyclonal anti-phospho- mammalian target of rapamycin; p38 MAPK: mitogen activated protein kinase; NF-kB: Nuclear factor kappa B; TNF-a: tumor necrosis factor-a; CRP: C-reactive protein; TAC: total antioxidant capacity; SOD: Superoxide dismutase; FRAP: ferric reducing ability of plasma; CAT: catalase; HLA: human leukocyte antigen; CD11b:(integrin αM); AMPK:AMP activated protein kinase; MDA: malondialdehyde; p70S6K: 70 kDa ribosomal protein S6 kinase, PKB: protein kinase B….”

Reviewer 2 Report

A possible role of NAC in the prevention of oxidative stress through mitohormesis and sarcohormesis induction in athletes is presented. Several points as indicated below need to be addressed by authors to improve the quality of the manuscript.

In the manuscript, the terms reactive oxidants and reactive oxygen species (ROS) are used. Please, explain the difference between the terms or unify the terms. I am not familiar with the “reactive oxidants” term.

Please explain the definition of redox hormesis. Did you mean the redox adaptation processes?

Line 24: This narrative review aims to re-evaluate the metabolic effects of NAD on exercise-induced oxidative stress… - What is NAD? Did you mean NAD+ (Nicotinamide adenine dinucleotide)?

Line 60: Is glutathione endogenous or exogenous antioxidant? Please clarify.   

195: Please explain how the skeletal adaptive response can be improved by diet (what kind of diet?) and antioxidants (what specific antioxidants)?

Line 205: Various antioxidant supplements… - please explain what specific antioxidants

Line 222: For instance, recent evidence has revealed that NF-KB activation may be associated with cellular glutathione concentrations [94]. – 94. is not a recent reference (1995)!

Mihm, S.; Galter, D.; Dröge, W. Modulation of transcription factor NFχB activity by intracellular glutathione levels and by variations of the extracellular cysteine supply. FASEB J.1995, 9, 246–620252, doi:10.1096/fasebj.9.2.7781927.

Paragraph 4.1.

Line 5:  NAC supplementation has extensively studied the effects of NAC on body antioxidant defence in different populations using a variety of doses, periods, duration, and exercise protocols [93]. “doses, periods, duration” - which of these relates to NAC and which to exercise type?

Line 70: NAC pre-supplementation generally acts by enhancing endogenous antioxidant capacity. Please add an explanation of this phenomenon.    Line 101: “antioxidant paradox” should be explained in detail.  Paragraph 4.3. Uncertainties and Future Perspectives on N- Acetylcysteine This paragraph is not well written. Instead of making a synthesis of what has been written so far, the intravenous NAC infusion is problematized. Additionally, Future Perspectives on N- Acetylcysteine is missing.  Figure 1 is unclear and should be additionally explained in the main text.  

Table 1. Please explain what is the definition of »Redox-related Study Outcomes«. Explain how inflammation markers e.g. IL-6, are related to redox measurements/outcomes.

Please explain which studies from Table 1 confirmed the N-acetylcysteine induced mitohormesis or sarcohormesis in athletic population.

Besides, in the manuscript, many sentences are unclear and difficult to understand. Professional English proofreading and editing services should be performed.

For example:

Line 92: Mitohormesis contains an increasing number of mitochondria, increased endogenous antioxidant responses, referred to as mitochondrial biogenesis, and altered cytoprotective gene expressions – unclear sentence. Did you mean »Mitohormesis involves increasing numbers of mitochondria, increased endogenous antioxidant responses termed mitochondrial biogenesis, and altered cytoprotective gene expressions«?

Line 132: Exercise-mediated reactive oxidants create an enormous mechanical overload on skeletal muscle, thereby leading to trigger anti-inflammatory response orchestrated by neutrophils and induce adaptive signaling pathways within the muscle [37]. Unclear sentence. Did you mean: »Exercise-mediated reactive oxidants create an enormous mechanical overload on skeletal muscle, resulting in an anti-inflammatory response orchestrated by neutrophils and induce adaptive signalling pathways within the muscle [37].«?

Line 134: Additionally, numerous myokines such as interleukin (IL)-6 and IL-10 start to synthesis in response to exercise-mediated adaptive- and hypertrophic signaling [38]. – Unclear sentence. Did you mean: »Additionally, numerous myokines such as interleukin (IL)-6 and IL -10 start to be synthesized in response to exercise-mediated adaptive and hypertrophic signals [38].«?

Line 199: NAC has been using various preclinical and clinical studies (Table 1). Unclear sentence. Did you mean: »NAC has been used in various preclinical and clinical studies (Table 1).«?    

Line: 201: please find a better expression for »powerful« antioxidant properties   

Line 33: One study drawing attention to a substantial point showed that previous glutathione levels before supplementation matter on NAC possible effects after a whole-body exercise [74]. Unclear sentence.

  Etc.          

Author Response

Answers to Reviewer 2

Many thanks for your valuable contributions and suggests. We really appreciate that you valued positively our work. We revised the paper based on your suggestions. All revisions are listed below;

  • In the manuscript, the terms reactive oxidants and reactive oxygen species (ROS) are used. Please, explain the difference between the terms or unify the terms. I am not familiar with the “reactive oxidants” term.
  • Answer: Based on your suggestion, we replaced the term "reactive oxidant" in the manuscript with the term "reactive oxygen species".

  • Please explain the definition of redox hormesis. Did you mean the redoxadaptation processes?
  • Answer: We truly mean the redox adaptation processes. We replaced the term “redox hormesis” in Line 180 with the term “the redox adaptation processes”.

  • Line 24: This narrative review aims to re-evaluate the metabolic effects of NAD on exercise-induced oxidative stress… - What is NAD? Did you mean NAD+ (Nicotinamide adenine dinucleotide)?
  • Answer: Thank you for your great attention. We wrote NAD by mistake. We replaced the term to “NAC”.

  • Line 60: Is glutathione endogenous or exogenous antioxidant? Please clarify.   
  • Answer: Glutathione (GSH) is one of the most powerful endogenous antioxidants that can be synthesized primarily in the liver. In addition, it can be obtained exogenously from diet or supplements. In line 60, we mean that NAC is considered one of the most promising antioxidants compared to other antioxidant supplements such as glutathione, vitamin E, and C.

We revised the paragraph as follows: “…..NAC is considered one of the most promising antioxidant substances compared to other antioxidant supplements such as glutathione, vitamin E and C [13]. Glutathione (GSH) is one of the most powerful endogenous antioxidants that can be synthesized primarily in the liver [14]. In addition, it can be obtained exogenously from diet or supplements [15]…..”

  • Line 195: Please explain how the skeletal adaptive response can be improved by diet (what kind of diet?) and antioxidants (what specific antioxidants)?
  • Answer: On your suggestion, we extended the explanation as follows: “….Therefore, it is essential to improve the skeletal adaptive response using endogenous adaptations by upregulating cytoprotective signal transduction cascades [58], or exogenous supplements with antioxidants such as vitamin C, E, [46] polyphenols including flavanols and anthocyanidins [59], or a balanced diet rich in antioxidants (e.g.; a diet rich in vegetables, fruits, chocolate, nuts, and their products) [60,61]….”

  • Line 205: Various antioxidant supplements… - please explain what specific antioxidants
  • Answer: On your suggestion, we revised the sentence as follows: “….Various antioxidant supplements such as vitamin E, C and a-lipoic acid reduce exercise-induced ROS and muscle fatigue and act as a reactive oxygen scavenger, improving recovery and anti-inflammatory response [65];….”

  • Line 222: For instance, recent evidence has revealed that NF-KB activation may be associated with cellular glutathione concentrations [94]. – 94. is not a recent reference (1995)!
  • Answer: Thank you for your valuable attention. We amended the sentence as follows: “…..For instance, Mihm et al. [76] has revealed that NF-KB activation may be associated with cellular glutathione concentrations…..”

  • Paragraph 4.1. Line 5:  NAC supplementation has extensively studied the effects of NAC on body antioxidant defense in different populations using a variety of doses, periods, duration, and exercise protocols [93]. “doses, periods, duration” - which of these relates to NAC and which to exercise type?
  • Answer: Many thanks for your valuable contribution. Here, we wanted to highlight that many studies have been performed to determine the effects of NAC supplementation on antioxidant defense. However; the duration and dose of NAC applied in studies, as well as the exercise severity, varies between studies. We rewrote the sentences as follows: “….Several studies have been conducted on different populations to determine the effects of NAC supplementation on antioxidant defense. However; in addition to the duration and dose of NAC supplementation, the exercise intensity, such as muscle-damaging, fatigue-inducing or high-intensity exercise, also varied between studies.…..”

  • Line 70: NAC pre-supplementation generally acts by enhancing endogenous antioxidant capacity. Please add an explanation of this phenomenon.    
  • Answer: On your suggestion, we revised the sentences as follows: ”…. These studies indicate that NAC pre-supplementation generally acts to increase endogenous antioxidant capacity, either directly by fighting elevated oxidants or indirectly by increasing glutathione synthesis or by inhibiting the rise of exercise-induced oxidants…..”

  • Line 101: “antioxidant paradox” should be explained in detail.  
  • Answer: We gave more detail on the term “antioxidant paradox” as follows: “….This term implies that although consumption of dietary antioxidants may provide beneficial effects in combating ROS to alleviate oxidative damage, high-dose antioxidant supplements can have detrimental consequences. One example is that oxidative radical damage may develop in both those who consume less than the recommended amount of vitamin C and those who take excessive vitamin C supplements [105]. ….”

  • Paragraph 4.3. Uncertainties and Future Perspectives on N- AcetylcysteineThis paragraph is not well written. Instead of making a synthesis of what has been written so far, the intravenous NAC infusion is problematized. Additionally, Future Perspectives on N- Acetylcysteine is missing.  

Answer: Thank you for your great contribution. In this section, we mainly wanted to highlight a few ambiguous points regarding the use of NAC as an antioxidant agent. Therefore, we changed the title of this paragraph to "Uncertainties regarding the use of N-Acetylcysteine as an antioxidant supplement". We revised the content of the conclusion part and included our suggestions and perspectives for future work here.

  • Figure 1 is unclear and should be additionally explained in the main text.  
  • Answer: Many thanks for your contribution. We added an explanatory paragraph for Figure 1 to the manuscript as follows:”… Figure 1 summarizes the potential effects of NAC on mito- and sarcohormesis under exercise-induced oxidative stress. Exercise at different intensities, such as muscle-damaging exercise, fatigue-inducing exercise, high-intensity and repetitive interval exercises, cause excessive accumulation of ROS in the body. NAC supplementation can provide several advantages for maintaining mitohormesis by increasing endogenous antioxidant defense, suppressing pro-oxidants, and upregulating cytoprotective adaptations. It can also support sarcohormesis maintenance by upregulating cytoprotective adaptations, regulating myomiRs-mediated regulation, regulating insulin uptake / insulin sensitivity, reducing muscle fatigue / pain, and regulating plasma K+ levels.….”.

  • Additionally, we added a Figure legend to make the figure more understandable. “…Figure 1. Potential effects of NAC on mito- and sarcohormesis under exercise-induced oxidative stress. The potential effects of NAC supplementation may vary according to the exercise severity. Therefore, NAC supplementation has been studied on certain exercise intensities including muscle-damaging exercise such as eccentric and broken bronco, fatigue inducing exercise, and high-intensity or repeated bouts of intermittent exercise. Each exercise intensity is differently colored and shaped to represent changes in metabolic processes. ­, ¯, and ´ represent the alterations caused by muscle damaging exercise (e.g.; eccentric, broken bronco). ­, ¯, and ´ represent the alterations caused by fatigue-inducing exercise (e.g.; intense or submaximal prolonged). ­, ¯, and ´ represent the alterations caused by high-intensity or repeated bouts of intermittent exercise. For example, NAC supplementation has showed an enhancing effect of NF-KB activity after fatigue-inducing exercise, while reducing plasma NF-KB activation after muscle-damaging exercise. Abbreviations: NAC: N-acetylcysteine; PC: Protein carbonyls; GSH: reduced glutathione; GPX: glutathione peroxidase; SOD: superoxide dismutase; NADPH: Nicotinamide adenine dinucleotide phosphate; CR: Creatine kinase; GSSG: oxidized glutathione; 3MH: 3-methylhistidine; IL-6: Interleukine-6; TGSH: total glutathione; MCP: Monocyte chemotactic protein; PGC-1a: Peroxisome proliferator-activated receptor coactivator-1a, Cys: cysteine, TBARS: Thiobarbituric acid-reactive substances; PSC: peroxyl radical scavenging capacity; mTOR: polyclonal anti-phospho- mammalian target of rapamycin; p38 MAPK: mitogen activated protein kinase; NF-kB: Nuclear factor kappa B; TNF-a: tumor necrosis factor-a; CRP: C-reactive protein; TAC: total antioxidant capacity; SOD: Superoxide dismutase; FRAP: ferric reducing ability of plasma; CAT: catalase; HLA: human leukocyte antigen; CD11b:(integrin αM); AMPK:AMP activated protein kinase; MDA: malondialdehyde; p70S6K: 70 kDa ribosomal protein S6 kinase, PKB: protein kinase B….”

  • Table 1. Please explain what is the definition of »Redox-related Study Outcomes«. Explain how inflammation markers e.g. IL-6, are related to redox measurements/ outcomes.
  • Answer: Here, we would like to emphasize the study outcomes related to redox status. We replaced the term “Redox-related Study Outcomes” to “Study outcomes”.

  • NAC has been shown to alter plasma cytokine concentrations after exercise. We added this explanation in the manuscript as follows: “…..These studies indicate that NAC pre-supplementation generally acts to increase endogenous antioxidant capacity, either directly by fighting elevated oxidants and attenuating proinflammatory cytokines, or indirectly by increasing glutathione synthesis or by inhibiting the rise of exercise-induced oxidants…..” (line 84-87).

  • Also, we explained the relationship in line 100-106 as follows: “….NAC supplementation may not alter biomarkers related to mitochondrial biogenesis after acute exercise [68,74]. One study related to metabolic adaptation showed that supplementing NAC at a dose of 1200 mg for nine days before a fatigue-inducing cycling exercise in well-trained triathletes (1) enhanced plasma total antioxidant capacity, (2) lessened pro-oxidant biomarkers determined by plasma TBARS and urinary F2t isoprostane levels, (3) attenuated inflammation measured by IL-6 and monocyte chemotactic protein 1, and (4) facilitated post-exercise NF-KB activation, thereby up-regulating exercise-induced redox alterations and adaptive process [74]…..” (line 101-106).

  • Please explain which studies from Table 1 confirmed the N-acetylcysteine induced mitohormesis or sarcohormesis in athletic population.
  • Answer: On your suggestion, we added an explanation to the conclusion part as follows:”….. In addition, NAC studies have conducted on recreationally trained individuals, endurance trained men, and athletes. Three studies investigating the effect of NAC on sarcohormesis in the athletic population did not confirm NAC-induced sarcohormesis in athletes [101–103]. Two of these studies argued that NAC supplementation had no effect on sarcohormesis [101,102], while one reported that it negatively affected ROS-induced metabolic adaptations in skeletal muscle [103]. In contrast, three studies investigating NAC-induced mitohormesis stated that NAC supplementation positively affected mitohormesis [74,100,101]. On the other hand, three out of four studies in endurance trained men supported NAC-induced sarcohormesis [17,96,97], while one claimed the opposite [98]. Additionally, it has been reported that NAC supplementation has no effect on mitohormesis in endurance trained men [17]. Considering all the findings, it appears that studies on NAC-induced mitohormesis and sarcohormesis are very few and suggest conflicting results, especially related to sarcohormesis. Furthermore, all studies in athletic population have been performed on men. Therefore, further studies need to evaluate the possible role of NAC in both mitochondria and muscles in redox regulation processes, and also perform on both gender.….” (line 352-365).

  • Besides, in the manuscript, many sentences are unclear and difficult to understand. Professional English proofreading and editing services should be performed.
  • Answer: Thank you for your kind contribution. We revised the sentences in line with your suggestions. Additionally, one of authors, whose native language is English, checked the article for proofreading.

  • Mitohormesis contains an increasing number of mitochondria, increased endogenous antioxidant responses, referred to as mitochondrial biogenesis, and altered cytoprotective gene expressions – unclear sentence. Did you mean »Mitohormesis involves increasing numbers of mitochondria, increased endogenous antioxidant responses termed mitochondrial biogenesis, and altered cytoprotective gene expressions«?
  • Answer: Line 92: Paper was amended accordingly as follows: “….Mitohormesis involves increasing numbers of mitochondria, increased endogenous antioxidant responses termed mitochondrial biogenesis, and altered cytoprotective gene expressions….”

  • Line 132: Exercise-mediated reactive oxidants create an enormous mechanical overload on skeletal muscle, thereby leading to trigger anti-inflammatory response orchestrated by neutrophils and induce adaptive signaling pathways within the muscle [37]. Unclear sentence. Did you mean: »Exercise-mediated reactive oxidants create an enormous mechanical overload on skeletal muscle, resulting in an anti-inflammatory response orchestrated by neutrophils and induce adaptive signaling pathways within the muscle [37].«?
  • Answer: Line 132: Paper was amended accordingly as follows: “……Exercise-mediated reactive oxidants create an enormous mechanical overload on skeletal muscle, resulting in an anti-inflammatory response orchestrated by neutrophils and induce adaptive signaling pathways within the muscle…..”

  • Line 134: Additionally, numerous myokines such as interleukin (IL)-6 and IL-10 start to synthesis in response to exercise-mediated adaptive- and hypertrophic signaling [38]. – Unclear sentence. Did you mean: »Additionally, numerous myokines such as interleukin (IL)-6 and IL -10 start to be synthesized in response to exercise-mediated adaptive and hypertrophic signals [38].«?
  • Answer: Line 134: Paper was amended accordingly as follows: “….Additionally, numerous myokines such as interleukin (IL)-6 and IL -10 start to be synthesized in response to exercise-mediated adaptive and hypertrophic signals….”

  • Line 199: NAC has been using various preclinical and clinical studies (Table 1). Unclear sentence. Did you mean: »NAC has been used in various preclinical and clinical studies (Table 1).«?    
  • Answer: Line 199: Paper was amended accordingly as follows: “…..NAC has been used in various preclinical and clinical studies…..”.

  • Line: 201: please find a better expression for »powerful« antioxidant properties 
  • Answer: We altered the term “powerful” with “effective”.

  • Line 33: One study drawing attention to a substantial point showed that previous glutathione levels before supplementation matter on NAC possible effects after a whole-body exercise [74]. Unclear sentence.
  • Answer: We rewrote the sentence as follows: “…..Importantly, one study highlighted that body glutathione levels prior to supplementation also had a substantial influence in evaluating the possible effects of NAC after whole-body exercise [74]……”.

Round 2

Reviewer 2 Report

I believe the manuscript has been significantly
improved and now warrants publication in Antioxidants.